

# Negligible influence of livestock contaminants and sampling system on ammonia measurements with cavity ring-down spectroscopy

Jesper Nørlem Kamp[1,2], Albarune Chowdhury[1], Anders Peter S. Adamsen[3] and Anders Feilberg[1]

[1]Department of Engineering, Aarhus University, Aarhus, 8000, Denmark
5  [2]Arctic Research Centre, Aarhus University, 8000 Aarhus, Denmark
[3]SEGES, 8200 Aarhus N, Denmark

*Correspondence to*: Anders Feilberg (AF@eng.au.dk)

**Abstract.** Pollution from ammonia ($NH_3$) is a widespread problem due to the effects on the environment and human health. The agricultural sector accounts for nearly all $NH_3$ emissions in Europe; thus, focus has especially been on $NH_3$ emissions 10  from this sector in recent years. We need abatement techniques to reduce $NH_3$ emissions, and in order to evaluate the techniques, there is a strong need for reliable $NH_3$ measurement methods. Photoacoustic spectroscopy (PAS) is often used to measure $NH_3$ concentrations, but recent discoveries show interference from compounds typically present in livestock production and during agricultural activities. We tested the performance of the Cavity Ring-Down Spectroscopy (CRDS) from Picarro as an alternative to PAS for filter effects and laboratory and field calibrations with standard gasses. Furthermore, 15  concentrations of ten volatile organic compounds (VOCs) where determined by Proton-Transfer-Reaction Mass Spectrometry (PTR-MS) to test the potential interference of these VOCs. Both laboratory and field calibrations show excellent linearity over a large dynamic range of $NH_3$ concentrations. The analyzer shows a small humidity effect of up to a few ppb in the extreme case of a nearly water saturated air stream. Besides, from the negligible humidity dependence, there is no interference from the tested VOCs. Overall, the CRDS system performs well with only negligible influences from other compounds.

20  **1 Introduction**

Ammonia ($NH_3$) is an important atmospheric pollutant as it has negative effects on ecosystems and human and animal health (Aneja et al., 2001; Davidson et al., 2005). Deposition of $NH_3$ can lead to eutrophication and acidification, which has negative effects on biodiversity (Sheppard et al., 2011). $NH_3$ is not a greenhouse gas (GHG), but it is a precursor for aerosol formation thereby influencing the global warming, furthermore different oxidation pathways of $NH_3$ produce nitrous oxide ($N_2O$) and 25  nitric oxide (NO) (Zhu et al., 2013). $N_2O$ is a very potent GHG and NO is involved in atmospheric reactions producing tropospheric ozone (Aneja et al., 2001). As a precursor for aerosols and thereby formation of particles in the atmosphere, $NH_3$ has harmful effects on human health (Aneja et al., 2001; Baek et al., 2004). Emissions of $NH_3$ have an impact on ecosystems, global warming and human health, thus it is important to measure $NH_3$ concentrations correctly.

Agricultural activities, mainly manure application and management, accounts for around 94% of $NH_3$ emissions in Europe 30  (Nielsen et al., 2017), and much control in the agricultural sector intend to limit these emissions. Reliable measurements in the



agricultural sector are highly important to give accurate estimates of $NH_3$ emissions in order to reduce the emissions e.g. by validation of technological improvements within the agricultural sector. Many agricultural studies use photoacoustic spectroscopy (PAS) (Poissant et al., 2005; Rom and Zhang, 2010; Saha et al., 2010; Wu et al., 2012; Zhang et al., 2005; Zong et al., 2014), but findings by Rosenstock et al. (2013) and Liu et al. (2018) show high interference on $NH_3$ measurements from

a variety of organic compounds including carboxylic acids and alcohols. Hassouna et al. (2013) report non-constant bias in the results from PAS measurements on $NH_3$ and $N_2O$ caused by organic compounds often present at agricultural sites, which make the PAS measurements uncertain in an agricultural setting. Other agricultural studies use CRDS to measure ammonia concentrations (Maasikmets et al., 2015; Sintermann et al., 2011).

It is challenging to measure $NH_3$ concentrations correctly due to its properties of high water solubility and polarity, which

cause adsorption on surfaces in the sampling system and within the instrument (Rom and Zhang, 2010; Shah et al., 2006; Vaittinen et al., 2014). This "sticky nature" of $NH_3$ causes delays in the measurements, giving longer response times (Rom and Zhang, 2010; Shah et al., 2006; Vaittinen et al., 2014). Furthermore, it is a challenge to measure $NH_3$ in livestock buildings, where dust and particles provide large surface areas for its adsorption in particulate filters. $NH_3$ adsorption can affect response time and accuracy of the analyzers, causing a time delay and measuring errors.

The air inside animal production buildings contains a variety of chemical compounds, relatively high water content and high densities of particulate matter. Several $NH_3$ analyzers are sensitive to water vapor and other gasses (Bobrutzki et al., 2010; Huszár et al., 2008; Ni and Heber, 2008; Rom and Zhang, 2010; Rosenstock et al., 2013). Such interference can introduce errors depending on the instrument used. Proper calibration before and during laboratory and field measurement can reduce the potential errors and improve the accuracy of the measurement system. Quantification of the errors can enhance the quality

of the $NH_3$ measurement data, which is essential to research, emission abatement and policy-making. There are varieties of $NH_3$ measurements methods, Bobrutzki et al. (2010) conduct a field inter-comparison of eleven atmospheric $NH_3$ measurement techniques including Cavity Ring-Down Spectroscopy (CRDS), and the results show a good overall agreement between the instruments on an hourly basis.

CRDS provides measurements in real time with high sensitivity, high selectivity and a fast response time. The CRDS analyzer

maintains high linearity, precision and accuracy over changing environmental conditions without the need for frequent calibration. The work of Martin et al. (2016) led to an improvement of water vapor interference calculations on Picarro's CRDS for $NH_3$ measurements. The scale factor error they discovered is approximately 2% of the absolute water concentration. Unfortunately, our Piccaro analyzer has not incorporated this upgraded correction, but the analyzer still corrects for H2O interference; thus, we expect this to have insignificant effect on the results.

We wish to address the interferences from VOCs further as the PAS technique seems to be inadequate for measurements in some agricultural environments (Hassouna et al., 2013; Liu et al., 2018; Zhao et al., 2012). A great number of VOCs are emitted in agricultural buildings from e.g. silage, manure and the animals (Feilberg et al., 2010; Hafner et al., 2013; Hansen et al., 2012; Ngwabie et al., 2008; Shaw et al., 2007; Yuan et al., 2017). The VOCs can potentially interfere with NH3 measurements as seen for PAS.



Our focus is on CRDS and we use the Picarro $NH_3/H_2O$ analyzer (Picarro Inc., Santa Clara, CA, USA) for all measurements of $NH_3$. The manufacturer states a lower detection limit below 0.5 ppb + 0.1% of reading (1s, 1σ) and below 0.03 ppb 0.1% of reading (5 min, 1σ) (Picarro, 2017). The response time is stated to be less than 30 seconds for 0 to 3 ppm and a recovery time from 10 to 0.2 ppb of less than 1 hour (Picarro, 2017). Our aim is to demonstrate the performance of the CRDS analyzer

for measurements of $NH_3$ gas concentration under laboratory and field conditions. Furthermore, to identify and quantify potential interfering compounds present in livestock buildings with state of the art Proton-Transfer-Reaction Mass Spectrometry (PTR-MS) to measure VOC concentration. To our knowledge no studies has focused on the interference from VOCs on CRDS measurements, which cause major concerns for measurements with PAS techniques (Hassouna et al., 2013; Liu et al., 2018; Zhao et al., 2012) . Due to the interferences present in PAS measurement, there is a need for a better alternative,

which is debated in the scientific society. Another issue with PAS measurements is the long response time of up to 25 minutes (Rom and Zhang, 2010), which lowers the time resolution of a study. $NH_3$ is underestimated by approximately 14% and 2% after 12.5 and 25 minutes, respectively (Rom and Zhang, 2010). Typical measurements in cattle barns takes places at multiple points, e.g. Rong et al. (2014) measure at 7 points in a dairy cow building and Ngwabie et al., (2009) measure at 11 points in a dairy cow barn. The cycle time for a typical setup in a barn would be in the order of 3-5 hours, which makes it impossible to

see most temporal variations with only 4-8 measurements per measurements point a day. Thus, the response time is a key parameter for equipment measuring at multiple points as done in livestock buildings. CRDS has a potential to be more accurate, precise and faster responding alternative to PAS.

This study aims to validate CRDS for measurements in the agricultural industry, thus we test for interference with a number of compounds typically present in pig houses and cattle farms where $NH_3$ concentration measurements are routine. Laboratory

tests include determination of the response parameters linearity, response time, influence of particulate filters and chemical interference. Field tests include determination of the response parameters linearity, response time and particulate filter effect. To test for the effects of VOC, we used PTR-MS, a powerful tool to measure selected VOCs and trace gasses in real time. Protonated water is a soft ionization source that protonates VOCs in a non-dissociative manner (de Gouw and Warneke, 2007; Lindinger et al., 1998). High selectivity and sensitivity are key characteristics of the method along with short response time

(de Gouw and Warneke, 2007). PTR-MS is ideal to quantify low concentrations of potentially interfering compounds in real time.

In this paper, we address the accuracy of CRDS in terms of interference from a range of VOCs normally present during livestock production and manure handling. CRDS is one of the online and not labor-intensive methods with the potential to measure $NH_3$ emissions. The focus on reduction of $NH_3$ emissions (e.g. Directive on the reduction of national emissions of

certain atmospheric pollutants (EU 2016-2284) (EC, 2016)) makes reliable and accurate measurement techniques essential.



## 2 Materials and Methods

### 2.1 Chemicals, reagents and gasses

We used the following chemicals during the experiments: 70 mM acetic acid (VWR int. S.A.S., Fontenay-sous-Bois, France), 27 mM 1-propanol (Merck KGaA, Darmstadt, Germany), 1.3 mM 2-propanol (Sigma-Aldrich Chemie GmbH, Steinheim,

Germany), 0.7 mM acetaldehyde (Sigma-Aldrich), 4 mM propionic acid (Alfa Aesar GmbH & Co KG, Karlsruhe, Germany), 0.8 mM acetone (Merck), 10 mM methanol (VWR), 2.2 mM 1-butanol (Merck), 69 mM ethanol (CCS Healthcare AB, Borlänge, Sweden) and 44 mM butanoic acid (Alfa Aesar). Deionized water dissolved the chemicals to the wanted concentrations.

We used the following gasses during the experiments: 101 ppm $NH_3$ (± 10%) in N2 calibration gas (AGA A/S, Copenhagen,

Denmark), pure (99.99%) CO2 (AGA), pure (99.99%) CH4 (AGA) and as zero air compressed air passed through a bed of silica gel and charcoal to remove water, ozone, hydrocarbons and other common contaminants. Mass flow controllers (MFCs) from the EL-FLOW (Bronkhorst High-Tech B.V., Ruurlo, Netherlands) series regulated all gas flows with an accuracy of ± 5%.

### 2.2 Interference

A Picarro $NH_3/H_2O$ analyzer model G2103 (Picarro Inc., Santa Clara, CA, USA) measured the $NH_3$ concentration continuously and a high sensitivity PTR-MS (Ionicon Analytik, Innsbruck, Austria) measured concentrations of different VOCs for the interference tests. The drift tube setting was 600 V, 2.1-2.2 mbar and 60°C, which yield an E/N of approximately 130 Td. Fragmentation of alcohols are normal in PTR-MS and we use the fragmentation of alcohols as described by Brown et al. (2010) to calculate the final concentration with all fragments taken into consideration.

One stream of clean air passed through the headspace air over an aqueous solution containing a single compound. Another stream diluted the outflow from the headspace. We changed the airflows to get different concentrations of the compound in the gas phase. The CRDS and PTR-MS received the diluted air streams.

### 2.3 Linearity, calibration and filter effect

We used a flow dilution system with zero air and $NH_3$ calibration gas (101 ppm) to test the linearity of the CRDS

measurements. $NH_3$ gas concentrations for the calibration were in the range from 0.20 to 16.8 ppm in the laboratory and from 0.27 to 20.0 ppm under field conditions. We performed the calibrations in the laboratory without external filters. Introduction of all gasses in the field was through a multi-position rotary valve (MPV, Cheminert low-pressure valve, model C25, VICI AG International, Schenkon, Switzerland) for 6 min while CRDS was in normal sampling mode. We performed a single point calibration in the field to test the system integrity and analyzer response time by introducing 7.8 ppm $NH_3$ calibration gas

directly into the sampling lines manually removed from their position. We tested the PTFE filters in the laboratory for $NH_3$



signals by connecting filters used for 2 weeks to a clean air supply under heating to maximum 75°C. Monitoring of the $NH_3$ signals continued until the concentration went below 5 ppb; see Table 1 for abbreviations and specifications of the used filters. We performed laboratory tests on the response time by switching between ambient air and 1.02 ppm $NH_3$ with the MPV without external filters attached. We also tested response time to a step change in NH3 concentration with different external

particulate filters attached. The concentrations were 0.203 and 10.01 ppm $NH_3$ with filters of different pore size made of PTFE, glass fiber and quartz. Table 1 shows the specifications of the filters.

## 2.4 Field testing

We conducted field tests in a cattle barn with natural ventilation located in central Jutland outside Viborg, Denmark. The cattle barn is 9 m high, 60 m long and 36 m wide and naturally ventilated. We measured $NH_3$ concentrations in the cattle building

with CRDS combined with a 10 port (P1-P10) MPV (C25-61800, VICI Valco Inst. Co. Inc., Texas, USA). Measurements were set up according to Rong et al (2014) and Wu et al. (2012). We considered the division into three 20 m sections inside the cattle barn to be representative of the animal-occupied zone of the barn. We sampled $NH_3$ concentrations from the three sections using PTFE tubes (inner diameter 6 mm, 20 m long) with 20 uniformly distributed sampling openings. The sampling points (SP) SP2, SP3 and SP4 were inside the building, with SP2 and SP4 on each of the end walls adjacent to the windows

placed 2.5 m above the floor. SP3 was just below the ridge opening in the middle of the building placed 9 m above the floor. SP1 and SP5 were outside as background measurements from two single points placed 5 m from the building sidewalls at 2.5 m height. The sample tubes were between 5 and 50 m long with heating cables attached to avoid condensation inside the tubing. Each sampling line had a secondary suction pump (flow rate of 6-7 L min$^{-1}$) with a PTFE membrane to generate a constant flow through the lines. A PTFE filter (0.20 μm pore size) removed airborne particulate matter from the sample air

before the sampling ports of the MPV. Replacement of filters was at last fortnightly. Measurement lasted 6 min for each sampling port with automatic switching, i.e. a measurement cycle was 30 min.

## 3 Results

### 3.1 Laboratory tests

The CRDS had a highly linear response ($R^2$=0.99998) to $NH_3$ concentrations over the dynamic range 0.20-16.8 ppm (Figure

1). The $NH_3$ standard calibration gas used for all calibrations had an accuracy of ± 10% stated by the manufacturer. The measured $NH_3$ concentrations in Figure 1 are averages of several hundred measurements and the standard deviations are indicators of stability. Figure 2 shows the result of a step change in concentration from clean air to 1.02 ppm and back to clean air, where the rise time to 95% concentration level was 13 s and the fall time to 5% concentration level was 19 s. Furthermore, Figure 3 shows response times to step changes to two concentrations (0.203 and 10.01 ppm) with different types of external

particulate filters. The response times varied for the different filter types with an average rise and fall time of 38 s and 52 s



(for 0.203 ppm), and 14 s and 12 s (for 10.01 ppm), respectively (Figure 3). Across all filter types, the response time was fastest for changes to the highest concentration, see details in Table A2.

Measurements on zero air over an hour gave a standard deviation on the $NH_3$ concentration of 0.115 ppb. This gives a limit of detection (LOD) of 0.35 ppb for three standard deviations and a limit of quantification (LOQ) of 1.15 ppb for ten standard
deviations, see Table 2.

We used pure deionized water to produce a range of different humidity without any contaminating compounds. Figure 4 shows the effects of the humidity changes on the $NH_3$ signal from relative humidity (RH) ranging from 6.3% to 98.6%. The response to the change in humidity is linear ($R^2 = 0.83$) with $NH_3$ measurements from 1.3 to 4.6 ppb over the given RH range.

Figure 5 shows the CRDS signals from $CO_2$ and $CH_4$, where random fluctuations in the low ppb level are present. There are
no apparent interferences from these two compounds. All measured $NH_3$ concentrations for both compounds are below the LOQ.

Figure 6 shows the interferences of ten different VOCs, where each row represent one compound. The third column shows a clear pattern for increased water vapor with VOC concentration as water is introduced with the VOCs. $NH_3$ concentration increased with increased water vapor for all compounds. The observed interferences were in the range from 0.5 to 5 ppb $NH_3$
equivalents at VOC concentrations from 6 to 8000 ppb. These VOC concentrations range from levels comparable to field conditions up to levels 1-2 orders of magnitude higher than at field conditions.

## 3.2 Field Tests

In the field, the CRDS also has a highly linear response ($R^2=0.9995$) in the concentration range 0.27-20.04 ppm, see Figure 7. Figure 8 shows the results of a single point field calibration of the system integrity and response time to a sudden change to
7.8 ppm with response times varying from 6 to 25 s. The calibration gas used in the measurements showed in Figure 7 and Figure 8 had an uncertainty of $\pm$ 10%.

Figure 9 shows the signals from 2 weeks old external particulate filters flushed with zero air. The concentration maximum varied between 25 and 38 ppb. The peak values are comparable to typical ambient laboratory concentrations ranging from 14-37 ppb; see Table A1. Vacuum pumps applied a gas flow rate of minimum 6 L min$^{-1}$ through the filter over the two-week
period, thus minimum 120 m$^3$ of air went through each filter.

## 4 Discussion

The CRDS analyzer had a linear response during both laboratory test and field validation (Figure 1 and Figure 7) in the range from approximately 0.2 to 20 ppm; $NH_3$ concentrations in livestock buildings are normally within this range (Heber et al., 2006; Koerkamp et al., 1998). This is in agreement with the manufacturer specifications that guarantees the range from 0 to
500 ppb, and with operational and optional expanded range up to 10 and 50 ppm, respectively (Picarro, 2017). The field calibration show excellent agreement with the standard gas concentrations. The standard gas had an uncertainty of $\pm$ 10%





according to the data sheet from the manufacturer, which may well explain the small offset (< 4%) from the obtained concentrations seen in Figure 8. The LOD found in the present study is comparable to the manufacturer's specifications for 1 seconds integration time, as seen in Table 2. The charcoal filter used might not clean the zero air generated completely compared to analytical standard gasses, thus we expect even lower LOD and LOQ by the use of analytical gas standards instead of air filtered by activated charcoal. The performance of a single point field calibration showed very good agreement with the expected concentrations as seen in Figure 8.

It is a requirement to have fast responding analyzers to understand the dynamic behavior and diurnal variations of $NH_3$ concentrations in animal buildings. Ni and Heber (2008) suggest a response time of less than 2 min to capture temporal $NH_3$ concentration variations. The CRDS shows sufficiently low response times under laboratory (Figure 2 and Table A2) and field conditions (Figure 8). These times are also comparable to < 30 s for responses to 3 ppm as reported by the manufacturer (Picarro, 2017). Furthermore, there are no clear changes in response time without the use of an external particulate matter filter. The tested filters vary in response time, but it is clear that the concentration difference in the step change was important where increasing concentration differences gave decreasing response times. The manufacturer reports rise- and fall times of approximately 16 seconds, which, compared to the present results, are very similar with some variations. The response to a change from 0 to 1.02 ppm gave 13 s and 19 s for the rise- and fall time, respectively (Fig. 3). Rise times with external particulate filters connected were 16 s, 6 s, 13 s, and 25 s, for SP1, SP2, SP3 and SP5, respectively. Adsorption to sampling material including filters explains the differences in response time. Response times are in general faster for higher concentration differences; see Fig. 4, as surface saturation is faster. The observed concentrations of $NH_3$ released from 2 weeks old particulate filters (Fig. 6) suggest that adsorption of $NH_3$ to the filter material, surfaces and walls were negligible. The levels released over 1 minute (< 50 ppb) should be compared to a filter exposure of ammonia of >100 ppb (ranging in to low ppm levels) over 2 weeks. These results indicate that the use of external filters gives satisfying response times and no problems with long-term adsorption of $NH_3$ on the filter material.

The gasses $CO_2$ and $CH_4$ are present in the atmosphere in relatively high concentration compared to other trace gasses, and animals produce $CO_2$ and $CH_4$, thus elevated concentrations are normal in animal houses. Over a large dynamic area, we observed little scatter and no interference for $CO_2$ and $CH_4$ on $NH_3$ measurements as seen in Figure 5. The mean concentration of both compounds are below the LOQ.

For the interference of single VOCs, it was as expected that the different dilutions prepared from clean dry air mixed with humid headspace air over a VOC solution gave a correlation between water vapor and VOC concentration. This was also the case as seen in the subplots in the third column of Figure 6 for all ten volatile compounds. Martin et al. (2016) observe an interference from water vapor on $NH_3$ measurements due to spectral line broadening, which the manufacturer corrects for in all models produced after the publication. Our Picarro analyzer from December 2014 does not make this extra correction, thus we expected a small water dependency for $NH_3$, which we indeed see in Figure 4 and Figure 6. Figure 4 shows the humidity effect on the CRDS signal generated from pure deionized water and reveals a small dependency for water vapor, which the improvements by Martin et al. (2016) potentially remove. Nonetheless, our results show up to 4.5 ppb $NH_3$ for a nearly water



saturated air stream with an absolute $H_2O$ concentration of approximately 1.1%. Thus, in the extreme case of low $NH_3$ concentrations (e.g. 100 pbb) and very humid air, a water vapor interference of up to 5 % of the $NH_3$ signal may be present, but under normal conditions, this is negligible.

The ten tested compounds are normally present during agricultural activities in sub-ppm levels (Copeland et al., 2012; Yuan
et al., 2017). We choose a concentration range that covers a large dynamic area and exceeds the normal maximum concentration in e.g. livestock buildings to obtain the potential maximum interference and we only observed very small water induced interferences. E.g., a 1-butanol concentration of 1.5 ppm gave a CRDS $NH_3$ concentration of 0.9 ppb, i.e. less than 0.06% of a 1:1 interference, which Liu et al. (2018) observe for several organic compounds when using photoacoustic $NH_3$ analysis. Overall, the difference between high and low concentration for a single VOC was approximately 1-2 ppb, except
acetic acid with a difference of nearly 4 ppb, but the differences in water vapor was different for the different compounds. Acetic acid, 2-propanol and propionic acids were the only compounds with absolute humidity above 1% as we used higher flow rates over the headspace to obtain the targeted concentrations. The very moist sample of acetic acid had a maximum of 4.6 ppb NH3, which is very low compared to e.g. concentrations in animal buildings, which typically range from < 1 to 20 ppm, but in extreme cases up 50 ppm (Heber et al., 2006; Koerkamp et al., 1998). Thus, errors on few ppb introduced by
humidity effects would have overall impact on the results. For the given setup, the interferences from water vapor were in the same order of magnitude as the LOQ of 1.15 ppb. For more than half of the VOCs, the $NH_3$ concentration falls below the LOQ for all or most measurement. This demonstrates a very low interference from the investigated VOCs.

Our tests of the Picarro CRDS showed great linearity during both laboratory and fieldwork. The rise- and fall times to concentrations changes were sufficiently low to measure temporal variations in $NH_3$ concentrations. Examinations of external
particulate filters lead to no clear recommendations for filter material, but all filter gave acceptable response times and only small amounts of $NH_3$ adsorption compared to background levels. We used PTR-MS to determine accurate VOC concentration for ten compounds and the compounds gave negligible interference on CRDS $NH_3$ measurements.

*Code and data availability*. Data and code are available upon request to the corresponding author.
*Author contributions*. Conceptualization, A.F., A.C. and J.K.; Methodology, A.F., A.C. and A.P.A.; Validation, J.K., A.F and A.C.; Formal Analysis, J.K and A.C..; Investigation, J.K. and A.C.; Resources, A.F. and A.P.A.; Data Curation, J.K. and A.C; Writing-Original Draft Preparation, J.K. and A.C.; Writing-Review & Editing, A.F., and A.P.A..; Visualization, J.K.; Supervision, A.F. and A.P.A.; Project Administration, A.F. and A.P.A.; Funding Acquisition, A.F. and A.P.A.

*Competing interests*. The authors declare no conflict of interest.

*Acknowledgements*. The authors thank laboratory technicians Heidi Grønbæk Christiansen and technician Peter Storegård Niels for their valuable help during the experimental part of the study. The authors thank the Danish Milk Levy Foundation for funding this research.

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

**Tables**

**Table 1. Specifications of tested particulate filters with abbreviations.**

| Filter material | Pore size (µm) | Thickness (mm) | Diameter (mm) | Porosity (%) | Filter code |
|---|---|---|---|---|---|
| PTFE[1] | 0.2 | 0.08 | 25 | 74 | PTFE 0.2 |
| PTFE[1] | 0.5 | 0.08 | 25 | 78 | PTFE 0.5 |
| PTFE[1] | 1.0 | 0.08 | 25 | 79 | PTFE 1.0 |
| PTFE[1] | 3.0 | 0.08 | 25 | 83 | PTFE 3.0 |
| PTFE[1] | 5.0 | 0.08 | 25 | - | PTFE 5.0 |
| Glass fiber | 0.6 | 0.21 | 25 | - | GA 55 |
| Glass fiber | 0.4 | 0.56 | 25 | - | GB 140 |
| Glass fiber | 0.8 | 0.74 | 25 | - | GA 200 |
| Quartz fiber | - | 0.38 | 25 | - | QR 100 |

[1] Polytetrafluoroethylene

**Table 2. Limit of detection and limit of quantification from a one-hour stable measurements on zero air.**

| | N | Mean | SD | LOD (3 x SD) | LOQ (10 x SD) |
|---|---|---|---|---|---|
| $H_2O$ [%] | 2065 | 0.082 | 0.0019 | 0.006 | 0.019 |
| $NH_3$ [ppb] | 2065 | 0.636 | 0.115 | 0.345 | 1.151 |





**Figures**

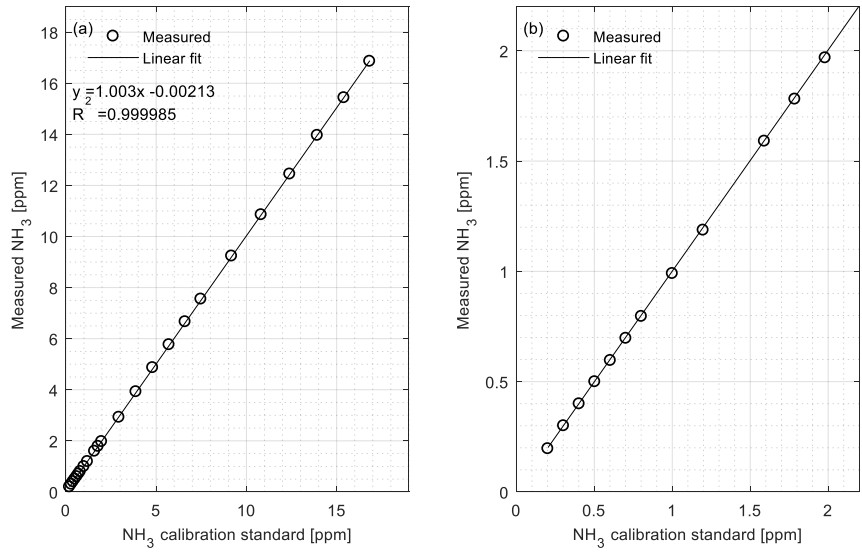

**Figure 1. (a) Calibration curve of CRDS from 0.20 to 16.8 ppm NH₃ conducted under laboratory conditions; (b) Calibration curve limited to 0 to 2 ppm. Symbols represent measured values, error bars the standard deviation, and the line is the fitted regression model.**

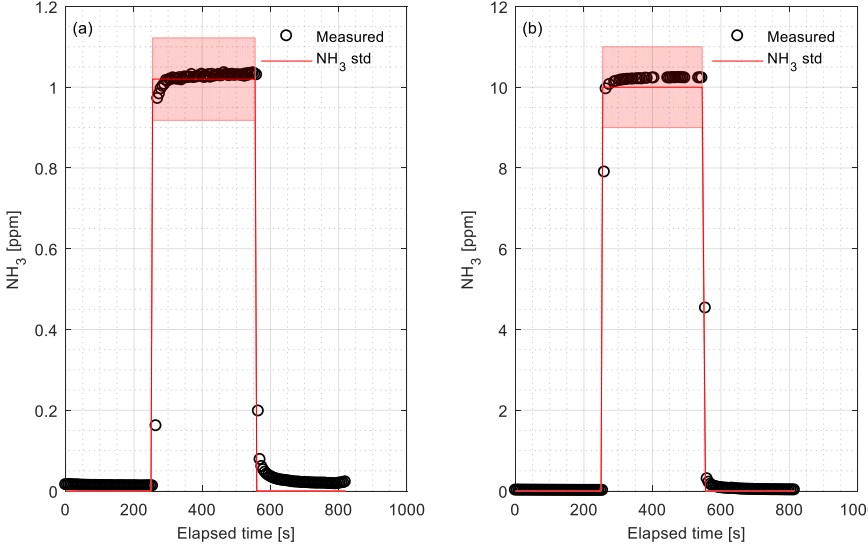

**Figure 2. (a) CRDS measurements during laboratory testing of the response to a step change in NH₃ to (a) 1.02 ppm with rise time (0% to 95% response) = 13 s and fall time (100% to 5% response) = 19 s; (b) 10.01 ppm with rise time = 8 s and fall time = 14 s. The red lines and areas represent the NH₃ standard gas concentration with 10% accuracy.**




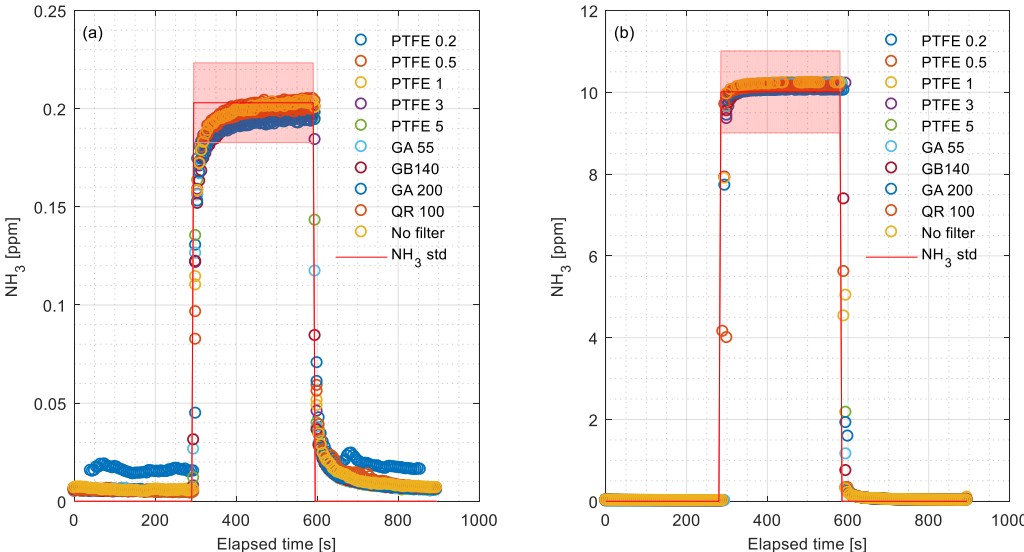

**Figure 3. The response to a step change in NH₃ at (a) 0.203 ppm and (b) 10.01 ppm with and without external inlet filters during laboratory testing. The red line and area represent the NH₃ standard gas concentration with with 10% accuracy. See the legend abbreviations in Table 1.**

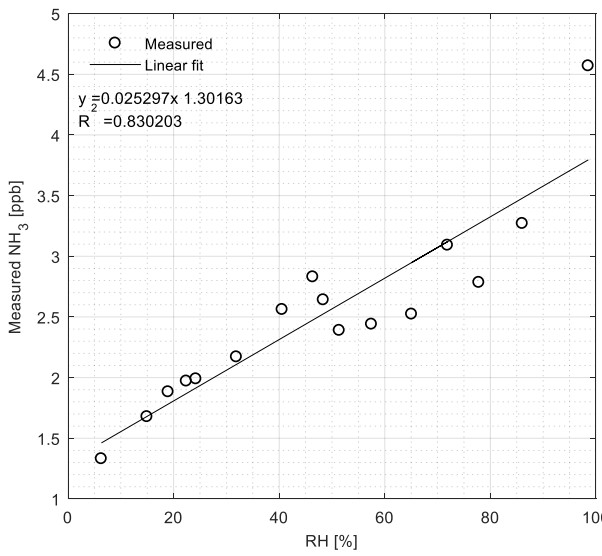

**Figure 4. CRDS signals of NH₃ (ppb) in zero air at different concentrations of water vapor, ranging from 6 to 99 % relative humidity (RH) at 22°C under laboratory conditions. Symbols represent measured values and the line is the fitted linear regression model.**





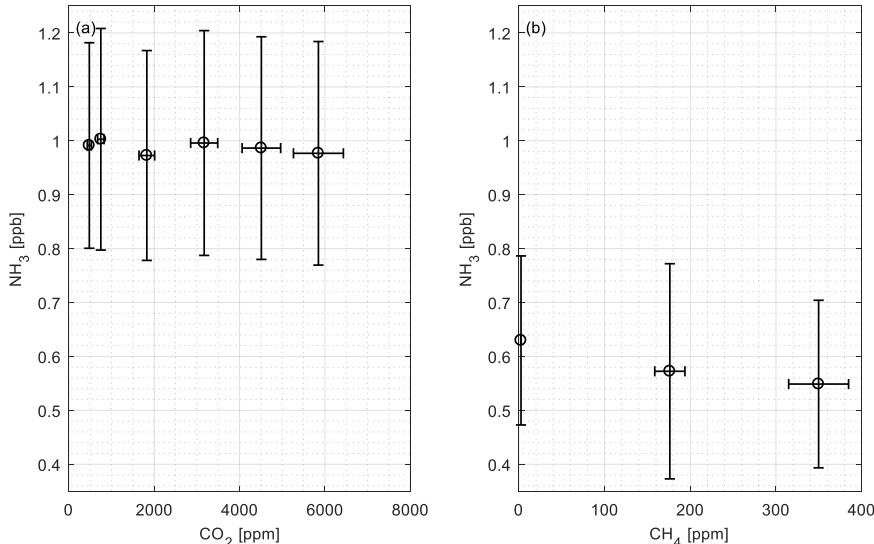

**Figure 5. CRDS signals of NH₃ in zero air response to various concentrations of (a) CO2 (480-5848 ppm) and (b) CH4 (2.42-350 ppm) under laboratory conditions. Symbols represent measured values and vertical and horizontal bars the standard deviation of the measurements.**

**1-Butanol interference**

**1-Propanol interference**

**2-Propanol interference**

**Acetaldehyde interference**

**Acetic Acid interference**



**Acetone interference**

**Butanoic Acid interference**

**Ethanol interference**

**Methanol interference**

**Propionic Acid interference**





**Figure 6. Interference of different organic compounds. Panels with subscript 1 show the CRDS signal to the compound. Panels with subscript 2 show CRDS signal to the absolute water content. Panels with subscript 3 show absolute water content and the concentration of the compound. The compounds are (a) 1-butanol; (b) 1-propanol; (c) 2-propanol; (d) acetaldehyde; (e) acetic acid; (f) acetone; (g) butanoic acid; (h) ethanol; (i) methanol; (j) propanoic acid.**

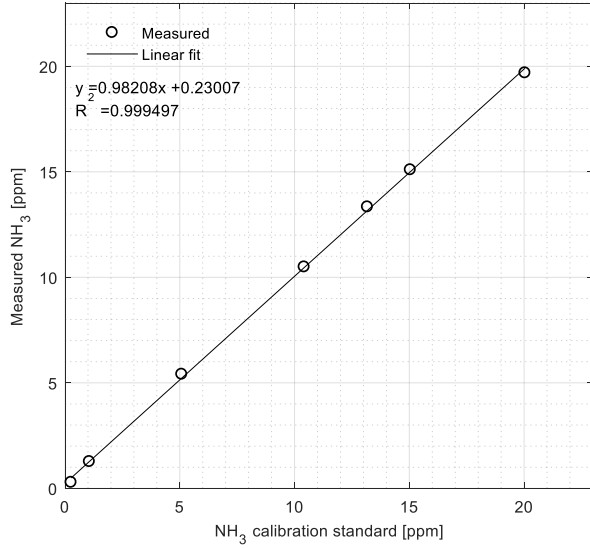

**Figure 7. Calibration curve of CRDS from 0.27 to 20.04 ppm NH₃ conducted under field conditions. Symbols represent measured values and the line is the fitted linear regression model.**





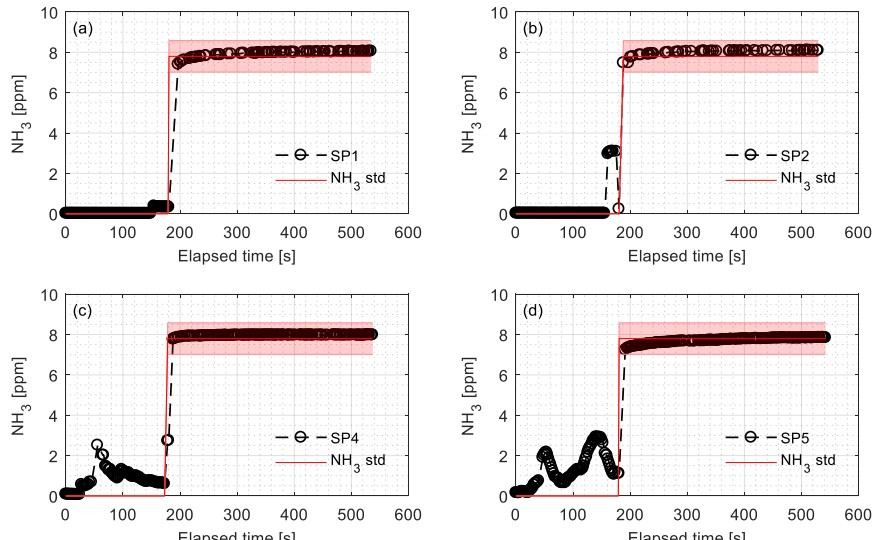

**Figure 8. Calibration of the NH₃ sampling and measurement system and associated response times of CRDS during field-testing. Introduction of 7.8 ppm NH₃ gas was at (a) SP1; (b) SP2; (c) SP4 and (d) SP5 while monitoring the NH₃ concentration at the outlet port connected to the analyzer. SP denotes sampling point. The rise times were 16 s, 6 s, 13 s, and 25 s, for SP1, SP2, SP3 and SP5, respectively. The reed line and area represent the NH₃ standard concentration with uncertainty.**

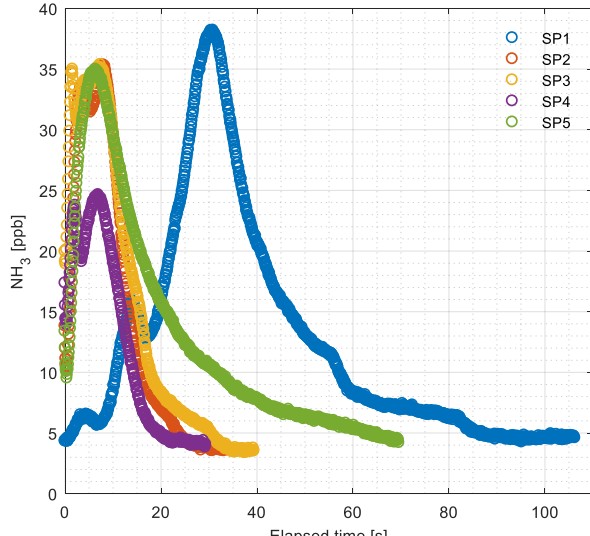

**Figure 9. Levels of NH₃ concentration in the 2-week old external particulate filters (PTFE, 0.20 µm pore size) measured by the CRDS in the laboratory. Filters collected from the field were installed before each sampling port. SP denotes sampling point.**



**Appendix A**

**Table A1. Measured concentrations of NH₃, CO₂ and CH₄ in laboratory zero air (n = number of data points; CV = coefficient of variation).**

| Gas | Concentration | Mean | CV (%) | Range (min-max) | Measurement duration (h) |
|-----|---------------|------|--------|-----------------|--------------------------|
| $NH_3$ | ppb | 1 | 43 | 0.1-5.5 | 19 |
| $CO_2$ | ppm | 480 | 5 | 460-545 | 19 |
| $CH_4$ | ppm | 2.4 | 10 | 1.7-3.2 | 19 |

5   **Table A2. This Rise (0% to 95% response time) and fall (100% to 5% response time) times (s) of the CRDS analyzer for measurements of NH3 concentrations (0.203 and 10.01 ppm) with or without external inlet particulate filters during laboratory testing. For filters detail, see Table 1.**

| Filter code | 0.203 ppm NH₃ | | 10.01 ppm NH₃ | |
|-------------|----------|----------|----------|----------|
| | Rise (s) | Fall (s) | Rise (s) | Fall (s) |
| No filter | 30 | 55 | 8 | 14 |
| PTFE 0.2 | 15 | 50 | 10 | 17 |
| PTFE 0.5 | 25 | 50 | 18 | 13 |
| PTFE 1.0 | 35 | 50 | 8 | 13 |
| PTFE 3.0 | 35 | 50 | 15 | 4 |
| PTFE 5.0 | 40 | 45 | 27 | 13 |
| GA 55 | 50 | 45 | 13 | 11 |
| GB 140 | 65 | 50 | 9 | 16 |
| GA 200 | 55 | 50 | 18 | 6 |
| QR 100 | 30 | 70 | 13 | 13 |
| Min | 15 | 45 | 8 | 4 |
| Max | 65 | 70 | 27 | 17 |
| Mean | 38 | 52 | 14 | 12 |