# Peer review of "Negligible influence of livestock contaminants and sampling system on ammonia measurements with cavity ring-down spectroscopy"

_Atmospheric Measurement Techniques, 2018_

## Referee Comment (RC1) · Anonymous Referee #1 · 8 Jan 2019

The manuscript describes field and laboratory experiment to test the influence of livestock contaminants (VOCS) and parts of the sampling system on Picarro-CRDS ammonia concentration measurements. The manuscript is well-written and the conclusions are clear. While the experiments are well described, I think that one or two important steps are missing which would strengthen the conclusions and show that the Picarro is the instrument to use under conditions with extremely high ammonia concentrations (>1 ppm). After some major corrections, (major as another test is needed, minor as I think it should not take too much time to perform and add to the manuscript).

< General comments and suggestions >

[Figure]

Main comments: 1. The authors mention that the VOC's can potentially interfere with Picarro-CRDS NH3 measurements, but what is the actual physical basis behind this statement? Does any of the VOC's have an absorption line near the two NH3 lines used in the Picarro instrument? (Such as the H2O lines that we can see in the water vapour case, shown in Martin et al., 2016) Why would any of the VOC's have an influence on NH3? Looking at the results in Figure 6, the bias in the NH3 concentrations follow directly from H2O, which was shown in the past by Martin et al., 2016. If we would remove the VOC sections from the manuscript, what remains is a non-conclusive test of the different filters and a response time test under extremely high concentrations with long inlet lines, which I think in itself and in its current form, is not original enough to warrant publication.

2. I would like to see the humidity test (and in extension the CO2/CH4 and VOC.. tests) under extremely high ammonia conditions. The author states the intent to use the Picarro to measure NH3 concentrations in animal production buildings. Concentrations inside such facilities are up to 4 orders higher than the usual atmospheric concentrations found in most (outside) field experiments (1-20 ppb, for example see Bobrutzki et al., 2010, where the range is 0.07 − 25 ppb for the CRDS). While the current tests show that one should definitely not use this Picarro-CRDS under normal atmospheric conditions (with a bias of the order of a few ppb) without an H2O correction, this does not have to be true inside the animal productions facility, where concentrations are of the order of a few ppm. The current results however do not show if the instrument bias changes under extremely high concentrations. If the authors can reproduce the few ppb bias even under extremely high nh3 concentrations (>1ppm) I would with confidence use this instrument inside such a facility. Furthermore, does temperature not influence the measurements? All lab experiments seem to have been done under a fixed temperature of 22 degrees, maybe vary this if possible?

General comments: 3. The manuscript describes conditions with concentrations up to an order or 3 higher than the usual atmospheric concentrations reported in other field

experiments. It might be good to add a plot/figure showing one or more time series of the typical concentrations and variations from hour to hour / day to day. This will give the reader a better grasp on the expected concentrations. It will also show if the 1.02-ppm and 10.01 ppm experimental air mixtures are sensible amplitudes to test the instrument responses for.

4. While the field and lab setups are well described it will be helpful for the reader to have a picture or schematic of the setup with the relevant parameters shown. (filter locations / inlet lines / inlet height).

5. What is the main reason that the PTR-MS was not used under field conditions? (p3, l20-21)

6. While the NH3 calibration gas is stated to have a concentration of 101ppm +-10%, I assume that the gas is still well mixed inside one bottle. If the same bottle was used for all tests, one can assume a constant bias. Looking at Figure 2 I would argue that the mixture is biased slightly high (as each of the response curves ends up at the fixed mixture value). Any variations in the signal follow from the Picarro system and local conditions.

7. Response time is defined as the time it takes to go from 100 – 5% and 0 to 95%. Is this 95% of the final level or of the expected concentration of the mixture? Why not fit an exponential function to the rise and fall sections of the graph? From those fits one can derive the e-folding time of the instrument instead of the to me less informative 95% check.

If possible adjust the figures to show the 20 or so seconds of interest. Split figure 2 into 4 sections. (a) and (b) covering the Rise and fall time of the 1.02 (more like 1.05) mixture, and (c) and (d) the rise and fall time of the 10.01 ppm (more like 10.25 ppm) mixture. 8. The response times of SP1 – SP5 seems to mostly to be related to the length of the inlet line and the height that the inlet is positioned at (P7, l16). What are the actual lengths of the inlet lines? The longest fall response time was

found to be ∼70s under lab conditions. What is the maximum instrument response time found under field conditions? 9. Figure 3 shows a variety of starting and final NH3 concentrations for each of the filters. What is causing the higher initial and final concentrations of PTFE 0.2 (or GA200) and the lower step change? 10. Figure 4: What is causing the high variation in the NH3 bias as a response to the humidity? Is there any possibility that the zero air is not completely zero? 11. Figure 6. While interesting to see NH3 to VOC, H2O and H2O to VOC I think you can show all of this in one figure. The authors show in figure 4 that NH3 has a relation to H2O. If possible, normalize the figure by diving NH3 over the fitted value. Show that the VOC's indeed have no effect. Furthermore, similarly as mentioned in point 2, test the bias under extremely high NH3 concentrations. 12. Table A1 shows a range of 0.1 – 5.5 ppb for NH3, what is causing these variations? Does the humidity change? And even if it does, why do we find 5.5 ppb for NH3 when the max bias was shown to be ∼5ppb under 100% humidity conditions. 13. I am missing a description of the measurement principle of the Picarro-CRDS. If possible add a description to section 2.2.

< technical corrections >

Throughout the manuscript, there are a number of grammatical errors. If possible, let someone with English as the mother tongue edit the document before final submission.

P2,l22 : Quantify, good is objective. Also Bobrutzki et al., 2010, showed an inter-comparison between instruments for concentrations around 1-25 ppb, not ppm's. P3,l16 : "The CRDS", change throughout the document

---

## Referee Comment (RC2) · Anonymous Referee #2 · 23 Jan 2019

General comment According to the manuscript, the CDRS was calibrated under laboratory and field conditions. Due to possible interference of other compounds (water, dust, temperature) of between compounds (NH3, CH4, CO2,...), it is important to compare the specifications and performance of the instrument under field conditions by comparing results of simultaneous measurements performed by using this instrument and a reference method (e.g. gas washing for NH3, gas chromatographie for CH4, N2O and CO2). This information is missing. Please add to the manuscript whether these measurements were performed (or not). If so, please report the results of the comparison. If not, please comment in the manuscript why this was not performed, and how this is going to be checked before using the instrument for real under field

conditions.

Other comments: In general, for a number of compounds the subscripts are not as subscript in the manuscript. Page 3, lines 12-15. These are results, not part or an introduction.
* * *

---

## Author Comment (AC1) · 4 Mar 2019

Answer to anonymous referee #1

> > > We appreciate the comments and suggestions from the anonymous referee that have helped to improve the manuscript.

The manuscript describes field and laboratory experiment to test the influence of live-stock contaminants (VOCS) and parts of the sampling system on Picarro-CRDS ammonia concentration measurements. The manuscript is well-written and the conclusions are clear. While the experiments are well described, I think that one or two important

steps are missing which would strengthen the conclusions and show that the Picarro is the instrument to use under conditions with extremely high ammonia concentrations (>1 ppm). After some major corrections, (major as another test is needed, minor as I think it should not take too much time to perform and add to the manuscript).

< General comments and suggestions >

Main comments:

1. The authors mention that the VOC's can potentially interfere with Picarro-CRDS NH3 measurements, but what is the actual physical basis behind this statement? Does any of the VOC's have an absorption line near the two NH3 lines used in the Picarro instrument? (Such as the H2O lines that we can see in the water vapour case, shown in Martin et al., 2016) Why would any of the VOC's have an influence on NH3? Looking at the results in Figure 6, the bias in the NH3 concentrations follow directly from H2O, which was shown in the past by Martin et al., 2016. If we would remove the VOC sections from the manuscript, what remains is a non-conclusive test of the different filters and a response time test under extremely high concentrations with long inlet lines, which I think in itself and in its current form, is not original enough to warrant publication.

> > > We agree that the physical basis behind this statement should be taken into consideration. We have used the HITRAN database for absorptions lines of the most important VOCs presented in this study. It can be seen that acetic acid has an absorption line close to ammonia and methanol does also have a line in the measurement range of the CRDS. This is solely from single-wavelength absorption, as line broadening is not taken into account. The following text and Figure 1 has been added to section 1: The absorptions lines of acetic acid and methanol found in the HITRAN database (Gordon et al., 2017) are in the same range as the ammonia lines used for measurements in the CRDS, see Figure 1 in the attached with the caption. This highlights the importance of the study as the absorption from VOCs can cause similar interference

as reported by Rosenstock et al. (2013) and Liu et al. (2018) for PAS.

The following has been added to section 2.2: The G2103 analyzer measure absorption from 6548.5 to 6549.2 cm-1 (Martin et al., 2016) and Figure 1 shows the absorption of some selected compounds in this range obtained from the HITRAN 2016 database (Gordon et al., 2017). The computed absorption lines in Figure 1 corresponds to 1% $H_2O$, 400 ppm $CO_2$, 100 ppb acetic acid, 100 ppb ethanol, and 100 ppb ammonia at 45°C and 140 Torr. Line broadening is not taken into account.

In addition, we would like to emphasize that the experiences with PAS (and lack of proper validation) have led to a strong need for documenting the limited interferences associated with CRDS. Thus, even if the theoretical contributions of VOC had been low, it would be important to carry out the tests performed in our study in order to convince the scientific community that CRDS is indeed a major improvement.

2. I would like to see the humidity test (and in extension the CO2/CH4 and VOC.. tests) under extremely high ammonia conditions. The author states the intent to use the Picarro to measure NH3 concentrations in animal production buildings. Concentrations inside such facilities are up to 4 orders higher than the usual atmospheric concentrations found in most (outside) field experiments (1-20 ppb, for example see Bobrutzki et al., 2010, where the range is 0.07 – 25 ppb for the CRDS). While the current tests show that one should definitely not use this Picarro-CRDS under normal atmospheric conditions (with a bias of the order of a few ppb) without an H2O correction, this does not have to be true inside the animal productions facility, where concentrations are of the order of a few ppm. The current results however do not show if the instrument bias changes under extremely high concentrations. If the authors can reproduce the few ppb bias even under extremely high nh3 concentrations (>1ppm) I would with confidence use this instrument inside such a facility. Furthermore, does temperature not influence the measurements? All lab experiments seem to have been done under a fixed temperature of 22 degrees, maybe vary this if possible?

> > > We have no expectations of a larger bias under higher NH3 concentrations as the contribution from water is expected to be independent of the ammonia concentration. It is not possible to separate the total measured ammonia concentration into a contribution from water and the pure ammonia signal. We have shown the extreme cases of water interference for this particular model and this is below 5 ppb, which will have negligible influence on ammonia concentrations above e.g. 1 ppm and still have a very low influence down to e.g. 100 ppb. Regarding the temperature, the cavity temperature and pressure is kept constant at all running times and will not run if these are not stable at the set point. Thus, changing the temperature would only change the equilibration time for absorption/desorption to the tubing and parts before the cavity. The wall absorption has been described by other e.g. (Shah et al., 2006; Vaittinen et al., 2014).

General comments:

3. The manuscript describes conditions with concentrations up to an order or 3 higher than the usual atmospheric concentrations reported in other field experiments. It might be good to add a plot/figure showing one or more time series of the typical concentrations and variations from hour to hour / day to day. This will give the reader a better grasp on the expected concentrations. It will also show if the 1.02-ppm and 10.01 ppm experimental air mixtures are sensible amplitudes to test the instrument responses for.

> > > Data from pig houses with finisher pigs can be added to show typical concentrations under different conditions over more than two months. Cycles were Room 1, Room 2, Room 3, Room 4, Background, etc. with 6 minutes measurements at each position (Hansen et al., submitted to Journal of Environmental Quality). Data presented in the figure is hourly mean values with concentrations up to 15 ppm; we suggest that it should be included in the appendix. Data from a dairy cattle barn can also be included, and the figure below shows hourly mean concentration over a week; we suggest that the figure is included in the appendix.

[Figure]

Figure A2 and Figure A3 are seen in the attached with captions

Added to section 3.1: This range is chosen from the expected concentration in a live-stock facility as Figure A1 and Figure A2 shows the hourly mean concentration of NH3 in four rooms with finisher pigs and a dairy cattle barn, respectively. The maximum concentration can exceed 15 ppm in the pig houses and 3 ppm in the cattle barn Added to section 4: . . . as seen in Figure A2 and Figure A3.

4. While the field and lab setups are well described it will be helpful for the reader to have a picture or schematic of the setup with the relevant parameters shown. (filter locations / inlet lines / inlet height).

> > > Figure A1 is added to the appendix with the caption seen in attached. The following text is added to section 2.4: The length of the sampling lines was approximately 5 m, 15 m, 35 m, 45 m, and 50 m for SP1, SP2, SP3, SP4, and SP5, respectively. See Figure A1 in the appendix.

5. What is the main reason that the PTR-MS was not used under field conditions? (p3, l20-21)

> > > Unfortunately, PTRMS data in field conditions are not available, but it would have been the logical next step to measure VOC in field conditions for comparison. We wanted to test for VOC interference in a controlled environment as there are a number of unknown factors and compounds present in the field. Therefore, this was our priority. Since VOC interferences are documented to be negligible for livestock emissions of NH3, we do not believe, however, that PTRMS measurements are critical for the conclusions. We have tested the VOC interference well beyond concentrations found in field conditions.

6. While the NH3 calibration gas is stated to have a concentration of 101ppm +-10%, I assume that the gas is still well mixed inside one bottle. If the same bottle was used for all tests, one can assume a constant bias. Looking at Figure 2 I would argue that

the mixture is biased slightly high (as each of the response curves ends up at the fixed mixture value). Any variations in the signal follow from the Picarro system and local conditions.

> > > We agree, the bias should be constant as the gas is well mixed in the bottle. The following is added to section 3.1: ... but the bias is considered constant as the gas is well mixed inside the bottle The following is added to section 4: ... but the system cause minor variations as the bias is considered constant.

7. Response time is defined as the time it takes to go from 100 – 5% and 0 to 95%. Is this 95% of the final level or of the expected concentration of the mixture? Why not fit an exponential function to the rise and fall sections of the graph? From those fits one can derive the e-folding time of the instrument instead of the to me less informative 95% check. If possible adjust the figures to show the 20 or so seconds of interest. Split figure 2 into 4 sections. (a) and (b) covering the Rise and fall time of the 1.02 (more like 1.05) mixture, and (c) and (d) the rise and fall time of the 10.01 ppm (more like 10.25 ppm) mixture.

> > > We have changed all rise- and fall times to be the e-folding time instead of using this 5% and 95% limits. This is achieved by fitting the rise and fall to an exponential function as suggested, which provides the time constant. All rise and fall times through-out the manuscript has been changed accordingly. Figure 2 (now Figure 3) has been changed, see the attached. Now, the relative concentration is used instead for a more direct comparison visual of the rise and fall time from the two different concentrations. As suggested, the figure has been split into two with focus on the rise and fall time of interest. The following is added to section 2.3: The response time for all experiments was found by fitting an exponential function to the step changes, which gave the e-folding time.

8. The response times of SP1 – SP5 seems to mostly to be related to the length of the inlet line and the height that the inlet is positioned at (P7, l16). What are the

actual lengths of the inlet lines? The longest fall response time was found to be ∼70s under lab conditions. What is the maximum instrument response time found under field conditions?

> > > The answer to general comment 4 addresses the lengths of the inlet lines. The response times have been changed after fitting it to a function to find the e-folding time as proposed, which makes it a bit more difficult to answer this question. Now, the response times are: SP1: 7.3 s (5 m) SP2: 3.0 s (15 m) SP4: 8.4 s (45 m) SP5: 5.9 s (50 m) Thus, there are no clear indications of a connection between response time and inlet line length.

9. Figure 3 shows a variety of starting and final NH3 concentrations for each of the filters. What is causing the higher initial and final concentrations of PTFE 0.2 (or GA200) and the lower step change?

> > > It is not completely clear why this was observed in a very few cases. One explanation is that contamination of the filters or tubes with background NH3 had occurred and that these low concentrations bled into the system before the addition of a standard gas. However, this does not affect the conclusions regarding the response times.

10. Figure 4: What is causing the high variation in the NH3 bias as a response to the humidity? Is there any possibility that the zero air is not completely zero?

> > > Firstly, it should be noted that these signals are very low, corresponding to a few ppb above the detection limit'. Thus, some variation in signals are to be expected. Secondly, the water vapor contributing the signals are created by bubbling air through water and the bursting of bubbles and evaporation of water may be subject to variations contributing to this pattern. If zero air had contained any significant amounts of ammonia, this would not be expected to vary much over the short time of the measurements and would have resulted in a uniform displacement of the regression line but not in more variation.

11. Figure 6. While interesting to see NH3 to VOC, H2O and H2O to VOC I think you can show all of this in one figure. The authors show in figure 4 that NH3 has a relation to H2O. If possible, normalize the figure by diving NH3 over the fitted value. Show that the VOC's indeed have no effect. Furthermore, similarly as mentioned in point 2, test the bias under extremely high NH3 concentrations.

> > > We agree. The relationship between NH3 and H2O has been used to correct the NH3 concentrations by subtraction the contribution from H2O. Thus, Figure 6 (changed to Figure 7) has been changed to contain only a single plot per compound with both un-corrected and corrected NH3 concentrations. The new Figure 7 is seen in the attached with the caption

12. Table A1 shows a range of 0.1 – 5.5 ppb for NH3, what is causing these variations? Does the humidity change? And even if it does, why do we find 5.5 ppb for NH3 when the max bias was shown to be 5ppb under 100% humidity conditions.

> > >We do observe some variation in the background signal fluctuating around an average of 1 ppb, which may be due to variable levels of residual ammonia in tubes etc. It is difficult to prove that the charcoal-filtered and dried air does not contain any ammonia above the detection limit, but the average level is close to the detection limit and some variation (reported as 43% in Table A1) can be expected. The value of 5.5 ppb is highest extreme and not representative. For this reason, we suggest removing the "Range" column in Table A1 as the CV% and mean better represents typical levels and variation.

13. I am missing a description of the measurement principle of the Picarro-CRDS. If possible add a description to section 2.2.

> > > Description of the measurement principle is added in section 2.2. "The opera-tional principle of CRDS relies on ring down time laser light. An air sample enters a cavity at low pressure (140 Torr) and laser light is pulsed into the cavity, where almost all light it is reflected by mirrors, which gives an effective path length of kilometers. A

small fraction of the light penetrates the mirrors to reach the detector and the intensity of the light is proportional to the concentration of target gas, as the target gas will absorb to light."

< technical corrections >

Throughout the manuscript, there are a number of grammatical errors. If possible, let someone with English as the mother tongue edit the document before final submission.

> > > We agree that the text needed a revision. Several of the authors have helped in the thorough proofreading of the manuscript.

P2,l22 : Quantify, good is objective. Also Bobrutzki et al., 2010, showed an intercomparison between instruments for concentrations around 1-25 ppb, not ppm's.

> > > Added to line 22: up to 120 ppb. Added to line 22: (R2>0.84)

P3,l16 : "The CRDS", change throughout the document

> > > Changed throughout the manuscript.
* * *
[Figure]

[Figure]

Figure 1 Simulated absorption spectrum from the HITRAN database for 1% H₂O (blue), 400 ppm CO₂ (green), 100 ppb acetic acid, 100 ppb ethanol, and 100 ppb NH₃ at 45°C and 187 mbar.

**Fig. 1.** Figure 1 Simulated absorption spectrum from the HITRAN database for 1% H2O (blue), 400 ppm CO2 (green), 100 ppb acetic acid, 100 ppb ethanol, and 100 ppb NH3 at 45°C and 187 mbar.

[Figure]

Figure A1 Schematic of the samplings point inside the cattle building. SP1 and SP5 were placed outside for background measurement at 2.5 m height. SP2 and SP4 were on the walls at 2.5 m height. SP3 was placed below the ridge at 9 m height. The lines were approximately 5 m, 15 m, 35 m, 45 m, and 50 m for SP1, SP2, SP3, SP4, and SP5, respectively.

**Fig. 2.** Figure A1 Schematic of the samplings point inside the cattle building. SP1 and SP5 were placed outside for background measurement at 2.5 m height. SP2 and SP4 were on the walls at 2.5 m height. SP3 wa

[Figure]

Figure A2 Hourly mean concentrations of NH₃ from four different rooms with finisher pigs, unpublished data.

**Fig. 3.** Figure A2 Hourly mean concentrations of NH3 from four different rooms with finisher pigs, unpublished data.

[Figure]

Figure A3 Hourly mean concentrations of NH₃ from a dairy cattle barn, unpublished data.

**Fig. 4.** Figure A3 Hourly mean concentrations of NH3 from a dairy cattle barn, unpublished data.

[Figure]

Figure 3. (a) Rise time and (b) fall time for the CRDS measurements normalized to final concentrations during laboratory testing of the response to a step change to 1.02 ppm (blue) and 10.01 ppm (green). The in NH₃ to 1.02 ppm with rise time (1/e) = 8.1 s and fall time (1/e) = 3.2 s; 10.01 ppm with rise time = 3.6 s and fall time = 4.8 s. The red lines and areas represent the NH₃ standard gas

**Fig. 5.** Figure 3. (a) Rise time and (b) fall time for the CRDS measurements normalized to final concentrations during laboratory testing of the response to a step change to 1.02 ppm (blue) and 10.01 ppm (gree

[Figure]

Figure 7. Interference of different organic compounds on the CRDS NH3 measurement. Blue markers indicate the original data and red markers indicate water corrected data from the regression showed in Figure 5. The compounds are (a) 1-butanol; (b) 1-propanol; (c) 2-propanol; (d) acetaldehyde; (e) acetic acid; (f) acetone; (g) butanoic acid; (h) ethanol; (i) methanol; (j) propanoic acid.

**Fig. 6.** Figure 7. Interference of different organic compounds on the CRDS NH3 measurement. Blue markers indicate the original data and red markers indicate water corrected data from the regression showed in F

---

## Author Comment (AC2) · 4 Mar 2019

> > > We appreciate the comments and suggestions from the anonymous referee that have helped to improve the manuscript.

General comment

According to the manuscript, the CDRS was calibrated under laboratory and field conditions. Due to possible interference of other compounds (water, dust, temperature) of between compounds (NH3, CH4, CO2,...), it is important to compare the specifications and performance of the instrument under field conditions by comparing results of si-

multaneous measurements performed by using this instrument and a reference method (e.g. gas washing for NH3, gas chromatographie for CH4, N2O and CO2). This information is missing. Please add to the manuscript whether these measurements were performed (or not). If so, please report the results of the comparison. If not, please comment in the manuscript why this was not performed, and how this is going to be checked before using the instrument for real under field conditions.

> > > We hypothesize that there are no interferences in the CRDS, which we test for and validate with laboratory tests of potential VOCs interference, CO2, CH4 and calibration with standard gas. We have not used a reference method as impingers or gas washing of ammonia as there are large variations in the concentration determination e.g. Misselbrook et al. (2005) report a coefficient of variance for absorptions flask of 21%. Furthermore, these methods are offline and rely on accumulation of ammonia over a long time span whereas the CRDS is running on a very different time scale, thus we do not find it suitable to use this as a reference method for the time-resolved concentration. The calibrations were conducted with a certified ammonia standard gas under a variety of conditions resembling realistic conditions, thus we find it sufficient as validation. Water, dust and temperature are mentioned as possible interfering compounds, and we have actually shown that water has a small interference in the model we are using, but this has been changed by water correction after the discoveries by Martin et al., (2016). The temperature is kept very stable at 45°C in the cavity, so temperature can only affect the sampling, which is also discussed in the introduction with the absorption of ammonia the walls and tubing. A filter removes dust mechanically, so it will never reach the cavity, and filter tests show a very small contribution from ammonia absorbed in the filter compared to the accumulated amount of ammonia flowing through the filter over a 2-week period.

Other comments:

In general, for a number of compounds the subscripts are not as subscript in the manuscript. Page 3, lines 12-15. These are results, not part or an introduction.

> > > All subscript are corrected to the right form throughout the manuscript. We think these results from other studies highlight the importance of response times, which was addressed in previous comments from the editor.

---

## Author Response (AR2)

**Associate Editor Decision: Publish subject to minor revisions (review by editor) (03 Apr 2019) by Folkert Boersma Comments to the Author:**
Your manuscript can be published in AMT after you make the following, mostly technical, corrections. Congratulations.

**Please consider shortening the introduction. There are repeated statements of what the aim of the study, that VOCs potentially interfere with the NH3 measurements etc.**
>>> *The introduction has been shortened and repeated statements have been removed.*

**One of the main conclusions is that "The results of this study clearly demonstrate the advantage of CRDS compared to the performance of the photoacoustic analyzers studied by Liu et al. (2019) for which severe VOC interferences on ammonia were observed." I suggest to phrase this a bit more cautious. Strictly, the advantage would only be proved if the two instruments were measuring NH3 side-by-side in a controlled setting with known H2O, VOCs etc. I think your work convincingly shows that the CRDS suffers from interference of these species to a limited extent, but it is not necessarily saying that PAS would fail if tested in the same Danish stable as you have done here. My suggestion is to nuance the statement. This can be done by pointing out that in your experiment the CRDS had only few interferences, whereas a PAS sensitivity test in comparable (?) circumstances, showed much more interferences from non-NH3 species.**
>>> *The conclusions have been nuanced and phrased more cautious:*
*"The results of this study clearly demonstrate the advantage of CRDS with only few and small interferences, whereas the performance of the photoacoustic analyzers under similar circumstances studied by Liu et al. (2019) showed much more interferences from non-NH$_3$ species."*

**P1, L10: "has been" --> have been**
>>> *Changed as suggested*

**P1, L31. Period missing after Nielsen reference.**
>>> *Period added.*

**P4, L26: Rong et al.**
>>> *Period added.*

**P5, L4: "expect for" --> except for?**
>>> *Changed to "except"*

**P6, L42: (100 pbb) --> 100 ppb?**
>>> *Changed to "NH3 concentrations of 100 pbb"*

**P7, L14: "errors on few ppb" --> errors of a few ppb would have a small (?) impact**

[revised manuscript text omitted]